# How Measurements "Affect" the Importance of Social Influences on Household's Photovoltaic Adoption—A German Case Study

**Ingo Kastner** [1,*]  **and Inga Wittenberg** [2]

1    Department of Environmental Psychology, Otto-von-Guericke Universität Magdeburg,
     39106 Magdeburg, Germany
2    Department of Personality and Social Psychology, Otto-von-Guericke Universität Magdeburg,
     39106 Magdeburg, Germany; inga.wittenberg@ovgu.de
*    Correspondence: ingo.kastner@ovgu.de; Tel.: +49-391-67-58476

**Abstract:** Investment determinants on residential photovoltaic (PV) in Germany were measured via questionnaire. The survey covered social influences in terms of injunctive and descriptive norms, and economic, ecological and autarkic motives for the investment. Descriptive norms were more relevant for the investment decisions than injunctive norms, but both were considerably less important than all of the other three investment motives. Additionally, we observed the actual distribution of PV systems in the participants' living area; we gathered the observation data on PV distribution from governmental databases. We found survey data on descriptive and objective norms and observation data to be unrelated. These findings indicate that multiple approaches are necessary to assess the relevance of social influences reliably.

**Keywords:** social influences; energy-relevant investment decisions; measurement problems

## 1. Introduction

Establishing an energy transition towards higher sustainability is one of the most pressing global challenges. Still, more than 80% of the worldwide energy is produced on fossil fuel basis, leading to $CO_2$ emissions and climate change. Nuclear sources cover another 5%. These sources do not cause $CO_2$ emissions but leave the unanswered question of permanent waste disposal. As compared to these sources, the share of renewable energies in worldwide energy production is rather low (about 14% [1]). The problems resulting from the current energy production, especially those related to climate change, are visible already, e.g., in terms of extreme weather, floods or changes in ecosystems. If a full energy transition is not initiated immediately, such developments are about to aggravate mostly hitting future generations [2].

One crucial element in stopping these developments lies in the adoption of "green" technologies, e.g., highly efficient equipment or systems producing renewable energies; these adoptions imply energy-relevant investments which are known to have a substantial positive impact on the environment [3]. Understanding how one decides in favor of (or against) an energy-relevant investment will be most helpful for designing policy measures. Unfortunately, there is not much research on energy-relevant investment decisions in social science yet. Only a small number of empirical studies have looked into the determinants of such investments, taking different theoretical perspectives and using various methods [4].

Most of the available studies conclude that making energy-relevant investment decisions is more likely if the decision makers expect personal benefits (e.g., cost savings and comfort increases [5,6]);

these egoistic motives seem to outweigh other determinants—such as eco-social motives or social influences. Recent analyses indicate that these findings may be biased, especially when it comes to social influences. In most studies, social influences are measured via direct questioning and appear to be of minor importance (see [4] for an overview). However, their decision relevance seems to increase if other measurement approaches (e.g., observations) are used [7,8]. The extent of these measurement biases is unclear because there are no studies offering systematic comparisons of different measurement approaches.

In questionnaire studies, it might also make a difference what constructs are used to measure social influences. They can, for instance, either be measured in terms of descriptive (i.e., what significant others are perceived to do [9–11]) or injunctive norms (i.e., what others are perceived to expect from the actors' perspective [9–11]). Some studies indicate that descriptive norms have a stronger impact on energy consumption behavior [12,13]. However, these studies are rather concerned with everyday behavior and have not yet been replicated for energy-relevant investment decisions.

The scope of this work is to properly capture the role of social influences in energy-relevant investment decisions. We will start with a brief outline of how social influences can be integrated into several well-established action models. We will provide more empirical findings showing that the decision relevance of social influences partly depends on the measurement method. Finally, we will present data from a study in which we compared different measurement methods of social influences. We will focus on PV adoption in households as one example for energy-relevant investment decisions.

## 2. Theory

### 2.1. Social Influences in Action Models

Social scientists widely agree that social influences affect behavior. Consequently, social influences are part of almost all relevant action models while their understanding may differ. The theory of planned behavior (TPB; [11]), the probably best-known action model in psychology, covers social influences in the form of subjective norms. Subjective norms are defined as the perceived social pressure to engage in a certain behavior. They result from the normative beliefs of what significant others expect one to do and the individual tendency to comply with these expectations. According to the TPB subjective norms do not affect behavior directly. They are mediated by the behavioral intention as well as attitudes and perceived behavior control. Meta-analyses of the TPB show that subjective norms have the least influence on behavioral intention compared to attitudes and perceived behavior control [14]

The norm-activation model (NAM; [15]) provides another approach to explain human behavior. Initially, this model was meant to describe "classic" pro-social behavior (e.g., altruism), but it has also been increasingly used to explain environmentally relevant behavior [16–18]. As the name implies, norms are central in this model with personal norms being the key construct. Personal norms are defined as a feeling of obligation to engage in a pro-social (e.g., a pro-environmental) behavior. Personal norms may affect behavior directly but need to be activated. A norm activation occurs if one realizes a social (e.g., environmental) problem, links the problem to his or her actual behavior, and recognizes that a behavioral change could contribute to the problem's solution. The NAM assumes that personal norms in part result from internalizing (pro-)social norms. Thus, the NAM takes a similar position as the TPB in capturing social influences; both models recognize that social influences do have some impact on behavior, but they are assumed to be mediated by other constructs.

Several efforts have been taken to integrate TPB, NAM and further theories into a comprehensive model. Two recent approaches are the stage model of self-regulated behavioral change [19] and the comprehensive action determination model [20]. Although these models have a somewhat different structure, they both also recognize social influences as one important driver of behavior, which is mediated by other constructs.

The diffusion of innovation theory (DOI; [21]) takes another view on social influences. Focusing on these influences, the DOI describes how an innovation (e.g., a new technology) spreads through society over time. Five adopter groups are categorized which successively adopt the innovation; the groups represent different parts of the population. Innovators are the first, rather small group to make an adoption (2.5% of the population), followed by early adopters (13.5%), the early majority (34%), the late majority (34%) and finally "laggards" (16%). These groups differ in several ways, e.g., education, economic and social status, willingness to take risks and technological affinity. Additionally, the groups have different positions when it comes to opinion leadership. Opinion leaders have the highest (social) influence on others. Thus, they play a key role in the adoption process—if they communicate an innovation to be favorable, it has better chances to spread over the whole population and vice versa. According to the DOI, early adopters hold the most influential opinion leadership. Consequently, they are a key group to target by policy measures fostering an adoption process.

### 2.2. Social Influences Are Often Underestimated

Several empirical studies show that people tend to underestimate the impact social influences have on their energy consumption [13,22]. In one study, Nolan, Schultz and Cialdini [12] measured social norms in two ways to reveal such biases. First, participants were asked directly how the expectations of significant others affected their energy conservation behavior (injunctive norm; e.g., [9,11]) and further motives (e.g., expected personal or ecological benefits). Second, they were asked if they think those significant others engaged in energy conservation behaviors (descriptive norm; e.g., [9–11]). Most interestingly, Nolan et al. [12] found that participants rated subjective norms to be least important as compared to other motives, while descriptive norms showed the highest impact on (self-reported and actual) energy consumption.

Culturally determined social desirability biases might favor the mismatch between the self-reported and actual relevance of social influences. Most empirical studies in this area have been conducted in individualistic countries (e.g., USA, Germany, the Netherlands or Norway; see [4]). In these cultures, self-determination is most important. Consequently, members of these cultures are expected to report highly independent decision making where others do not influence them (see [23,24] for more information on cultural differences).

### 2.3. Social Influences on Energy-Relevant Investment Decisions

Reflecting on this, we would like to note that the findings of the research presented so far may not be easily transferred to behavior in the domain of energy-relevant investments. Most psychological research on energy consumption behavior does not focus on energy-relevant investment decision but on everyday consumption behavior (see [20,25] for recent overviews). These two kinds of behavior are most different, though [26,27]. Energy-relevant investments, on the one hand, are rather complex actions. These investments are made rarely, have several consequences and thus involve high cognitive efforts. Everyday energy consumption behavior, on the other hand, frequently occurs where a single behavior has little consequences, and the behavior is often habitual.

Decades ago, Stern and Gardner [26] claimed that environmental psychologists should better focus on energy-relevant investment decisions—especially as they have a higher impact on energy consumption. However, there is still a substantial imbalance. Consequently, the validations of action models and the findings concerning the underestimations of social influences are mostly based on studies in the field of everyday energy consumption behavior. Given the differences between everyday consumption behavior and energy-relevant investments, it is unclear whether the findings can be transferred to the latter. Comparisons between different kinds of energy-relevant behaviors indicate that different determinants are relevant. In one study Black, Stern and Elworth [5] compared determinants of energy-relevant investment decisions and everyday consumption behaviors. Among several other factors, they measured social influences in terms of social norms. The authors found social influences being relevant for everyday consumption behavior and low-cost investment decisions (e.g., sealing

window leaks, weather-stripping) where they were mediated by personal norms; relationships between social influences and major energy-relevant investment decisions (e.g., insulation measures, purchasing more efficient furnaces) were not found.

The empirical work on energy-relevant investment decisions is not only fairly limited but also most heterogeneous [4,28]; the available studies differ in their theoretical backgrounds, the types of investigated investments, the assessed determinants and the methods used for measurements and data analyses. Social influences are mostly not treated as action models suggest, but rather as direct predecessors of behavior than being mediated by other constructs. In most studies, social influences were found to be of minor importance as compared to other factors (e.g., expected personal benefits; see [4] for an overview). That finding might partly arise from not considering the structures of action models suggesting rather indirect effects of social influences (see Section 2.1).

A closer look at the methods used across these studies also suggests that the relevance of social influences might be severely affected by the measurement [4]. In most studies, participants are asked retrospectively for their investment motives via direct questioning (e.g., questionnaires or interviews), where social influences mostly refer to injunctive norms (i.e., how the expectations of significant others affected the investment decision; see Section 2.2). Among most of these studies, injunctive norms were found to be rather unimportant as compared to other investment motives (see [4] for an overview).

In some recent studies, social influences have also been measured in terms of descriptive norms. The findings are not entirely conclusive. On the one hand, findings by Palm [29] indicate a rather high relevance of descriptive norms on PV adoption. On the other hand, Wolske, Stern and Dietz [30] compared a number of determinants, including injunctive and descriptive norms. They found the PV adoption intention to be more strongly affected by injunctive than by descriptive norms.

Some recent studies used different approaches than direct questions to measure social influence, suggesting a higher decision-relevance. One approach is to measure the decision makers' physical proximity to (visible) investments made by others. Among these studies, PV adoptions were found to increase with the number of PV installations in the decision makers' neighborhood [7,8,31,32] and to speed up the adoption decision [33]. These measurements can be interpreted as descriptive norm indicators as they depict the visible behaviors of others. Another more "indirect" measurement approach was integrated into discrete choice experiments on energy-relevant investment decisions. In these experiments, social influences were assessed in terms of investment recommendations given by trustworthy sources (e.g., energy consultants, peers [34,35]). Those recommendations were found to be an important investment determinant.

### 2.4. Research Agenda

Taken together, action models suggest that social influences are important for making energy-relevant investments while empirical data are ambiguous. A number of studies found social influences being rather unimportant as compared to other investment determinants (e.g., expected personal or ecological benefits). Some recent analyses indicate that these findings may be biased by methodological problems. So far, social influences have mostly been measured in terms of injunctive norms via direct questioning even though biases are likely if such approaches are used. Some studies using more "indirect" measurements (e.g., observations) indicate a higher relevance of social influences.

The extent of this measurement biases is unclear. Up to now, all studies in the field of energy-relevant investment decisions have relied on one approach to measure social influences, and a systematic comparison is widely missing. It is the objective of the present study to fill this gap.

We conducted a study on PV investments in German households. Social influences were measured in three ways: In an online survey, PV adopters were asked for the determinants of their investment decisions; among others, social influences were captured in terms of injunctive (I) and descriptive (II) norms. We compared both measurements to each other and to the other investment motives. We expected similar findings to the studies in the field of everyday energy consumption [12]:

**Hypothesis 1a:** *Descriptive norms are more relevant for energy-relevant investment decisions than injunctive norms.*

**Hypothesis 1b:** *Injunctive norms are less important for energy-relevant investment decisions than other investment motives. Descriptive norms are similarly important as other investment motives.*

In addition, we measured the number of PV systems installed in the neighborhood (III) as a further indicator of social influences. The observation data should be related to the questionnaire data. The relationship should be stronger between observation data and descriptive norms as both refer to the visible context.

**Hypothesis 2a:** *The observed number of PV systems in the living area correlates with the questionnaire data on social influences.*

**Hypothesis 2b:** *The correlation is stronger between observation data and descriptive norms than between observation data and injunctive norms.*

## 3. Materials and Methods

We collected survey and observation data in two steps, which are described in the following.

### 3.1. Survey

Participants used the online questionnaire between June and September 2015. The questionnaires were distributed among 15 PV related German web portals addressing households already owning a PV system and households interested in PV (e.g., PV web journals, associations, internet forums). We also used posts and newsletters to invite PV owners to participate, announcing a lottery to motivate participation. The measurement was conducted as part of a research program investigating energy-relevant behavior in households owning a PV system (see [36] for a detailed description). Thus, there was no room to investigate further energy-relevant investments that could be affected by social influences.

We measured social influences on PV investments in terms of descriptive and injunctive norms. The item design was based on Ajzen's TPB Questionnaire Construction Manual [10], which covers both types of norms. The items referred to two social groups, namely people who are important to the participant, and the neighborhood. For each group two items were included, one referring to the descriptive (i.e., the perceived PV installation made by the two groups) and one covering the injunctive norms (i.e., the perceived social support to make an own investment in PV). The items are presented in Table 1.

Three other motives for PV investment decisions were measured in order to compare the relevance of social influences to other determinants. We focused on motives that had proven to be relevant for residential PV investments before (e.g., [37,38]). More precisely, economic motives, ecological motives and autarkic motives were included; each motive was measured via two items (see Table 1).

All questions were measured on a 5-point scale (1 = Strongly disagree, 2 = disagree, 3 = neither agree nor disagree, 4 = agree, 5 = strongly agree).

**Table 1.** Overview of Survey Data.

| Investment Motives | M | SD |
|---|---|---|
| I purchased a photovoltaic (PV) system ... | | |
| Descriptive norms (r = 0.51) | 1.61 | 0.74 |
| because many people who are important to me owned one before I did. | 1.43 | 0.72 |
| because many people in my neighborhood owned one before I did. | 1.80 | 0.99 |
| Injunctive Norms (r = 0.69) | 1.25 | 0.46 |
| ... because people who are important to me expected me to do so. | 1.32 | 0.58 |
| ... because people in my neighborhood expected me to do so. | 1.19 | 0.42 |
| Economic motives ($r$ = 0.54) | 3.45 | 0.94 |

**Table 1.** *Cont.*

| Investment Motives | M | SD |
|---|---|---|
| ... because I expect it to be profitable. | 3.34 | 1.06 |
| ... to save money. | 3.56 | 1.08 |
| Ecological motives ($r = 0.75$) | 3.44 | 1.04 |
| ... to contribute to climate protection. | 3.56 | 1.14 |
| ... for ecological reasons. | 3.31 | 1.08 |
| Autarkic motives ($r = 0.64$) | 3.58 | 1.00 |
| ... because it gives me control over my energy provision. | 3.67 | 1.02 |
| ... to become independent of energy providers. | 3.50 | 1.19 |

Notes. 5-point scale from 1 = strongly disagree to 5 = strongly agree. All correlations were statistically significant at $p < 0.001$.

### 3.2. Observation Data

We assessed the number of PV systems per capita in the participants' living area. We defined living areas via zip codes; it was the most detailed resolution available. The data were calculated from two sources: the German Solar Energy Association who collected the information from the Electricity Grids Operators in Germany [39] provided data concerning PV distribution in Germany. The German Federal Office of Statistics [40] provided data concerning the population density in the participants' living areas.

### 3.3. Sample

A total of 247 data sets were gathered within the survey. Several data sets needed to be excluded due to missing values and incomplete information concerning the population density, leaving a sample of $n = 120$. All participating households installed their PV systems between 2009 and 2015. Most systems had been installed between 2011 and 2014 (94.1%; see Table 2). The average age of the 120 participants was 54 years ($M = 53.8$, $SD = 12.0$), almost all were male (99.2 %), and reported a high income (45.9% reported a net monthly household income > €3600) and a high education level (44.5% held a college university degree). Taken together, the sample is not representative for the German population—especially in terms of gender, income, and education. Nevertheless, the sample is similar to other samples of PV owners [41,42] and typical for the innovators and early adopters described in the DOI (see Section 2.1).

**Table 2.** Overview of sample characteristics.

| | Sample ($n = 120$) |
|---|---|
| **Age** | |
| M | 53.8 |
| SD | 12.0 |
| **Gender** | |
| Male | 99.2% |
| Female | 0.8% |
| **Education** | |
| University/College Degree | 44.5% |
| University Entrance Diploma | 12.6% |
| Secondary School Certificate | 15.1% |
| Other/not specified | 27.7% |
| **Net household income per month** | |
| Up to €1300 | 2.5% |
| €1301–€2600 | 20.8% |
| €2601–€3600 | 29.2% |
| €3601–€5000 | 31.7% |
| More than €5000 | 14.2% |
| Not specified | 1.7% |
| **Year of PV system installation** | |
| 2009 | 0.8% |
| 2010 | 0.0% |
| 2011 | 25.0% |
| 2012 | 18.3% |
| 2013 | 30.8% |
| 2014 | 20.0% |
| 2015 | 5.0% |

The observation data revealed a range of 0.0002 and 0.0771 PV systems per capita across the zip code areas. There were no two PV systems in the same zip code area. The data were gathered from the German Solar Energy Association and the German Federal Office of Statistics (see Methods).

## 4. Results

### 4.1. Investment Relevance of Social Influences and Other Investment Determinants

As our first step, we calculated scales for descriptive and injunctive norms, and for the economic, ecological and autarkic motives. We measured all norms and motives with two items; we calculated correlations to check scale reliability. All correlations were statistically significant showing moderate to strong relationships (descriptive norm: $r = 0.51$; injunctive norm: $r = 0.69$; economic motives: $r = 0.54$; ecological motives: $r = 0.75$; autarkic motives: $r = 0.64$).

The social norms' means indicate that descriptive and injunctive norms were both rather unimportant for the PV investment decisions, where descriptive norms were slightly more relevant (descriptive norm: $M = 1.61$; injunctive norm: $M = 1.25$). All other three investment motives were found to be more relevant. The autarkic motive was rated as most important ($M = 3.58$), closely followed by economic motives ($M = 3.45$) and ecological motives ($M = 3.44$). All descriptive data are shown in Table 1, which also contains descriptive statistics for each item.

We conducted mean comparisons between the two norms and the other motives in order to gain a better understanding of the differences between the investment determinants. Altogether, 10 mean comparisons were made using paired *t*-tests. The alpha level was adjusted to $\alpha = 0.005$. The results are shown in Table 3.

**Table 3.** Mean Comparisons of Investment Motives.

| Pair | $\Delta_M$ | $t$ | $df$ |
|---|---|---|---|
| **Descriptive Norm** *minus* | | | |
| Injunctive Norm | 0.36 | 5.67 * | 119 |
| Economic Motivation | −1.83 | −16.28 * | 117 |
| Ecological Motivation | −1.81 | −17.61 * | 117 |
| Autarkic Motivation | −1.98 | −16.89 * | 115 |
| **Injunctive Norm** *minus* | | | |
| Economic Motivation | −2.19 | −22.41 * | 117 |
| Ecological Motivation | −2.18 | −22.82 * | 117 |
| Autarkic Motivation | −2.34 | −23.56 * | 115 |
| **Economic Motivation** *minus* | | | |
| Ecological Motivation | 0.01 | 0.08 | 117 |
| Autarkic Motivation | −0.14 | −1.03 | 114 |
| **Ecological Motivation** *minus* | | | |
| Autarkic Motivation | −0.14 | −1.03 | 114 |

Notes. $\Delta_M$: mean differences between the motives (bold motive–unbold motive). * $p < 0.005$ (adjusted alpha level).

Mean comparisons confirm that descriptive and injunctive norms were less relevant for the PV investment decision than other motives. All mean differences between the norms and the other motives were statistically significant at the adjusted alpha level. Significant mean differences were also found between the two norm types, confirming the higher relevance of descriptive norms as compared to injunctive norms ($t(119) = 5.67$; $p = 0.000$). No statistically significant mean differences were found between economic, ecological and autarkic motives.

### 4.2. Relationships between Survey and Observation Data on Social Influences

We analyzed relationships between the survey data on descriptive and injunctive norms, and observational data (PV systems per capita) by calculating Pearson's correlational coefficient $r$. Again, separate analyses were run for descriptive and injunctive norms.

We found no statistically significant correlations between survey and observational data. As compared to each other, the relationship between descriptive norms and observational data ($r = 0.06$;

$p = 0.49$) was slightly stronger than the relationship between injunctive norms and observational data ($r = -0.16$; $p = 0.86$).

## 5. Discussion

### 5.1. Hypotheses Validation

Hypothesis 1a has been confirmed. Descriptive norms were more relevant for the PV adoption than injunctive norms; the difference between both determinants was statistically significant. Hypothesis 1b needs to be rejected. We only expected injunctive norms—but not descriptive norms—to be significantly less relevant than economic, ecological and autarkic motives; however, we found both injunctive and descriptive norms to be significantly less relevant.

In line with previous research, we found autarkic motives being the most influential determinant for energy-relevant investment decisions [4,37]. Autarkic motives could be interpreted as a counterpart to social influences. Especially in individualistic countries, independence is regarded as desirable while being influenced by others is not [23,24]. Consequently, the relevance of autarkic motives might be biased by social desirability, if they are measured in self-reports.

Hypotheses 2a and 2b need to be rejected as well, as we found no significant correlations between either descriptive norms and observation data, or injunctive norms and observation data. Especially for descriptive norms, we find these results most surprising. As descriptive norms should reflect the circumstances in the participants' living area, they should be strongly associated with the observation data. Naturally, self-reported descriptive norms can be biased to some degree [43], but we would not have expected a zero-correlation.

### 5.2. Limitations

A major shortcoming of our analyses is that our sample consisted of adopters only. The design would have been stronger if the results had been compared to a non-adopters sample. Such comparisons were not possible in our study: We gathered our data from German PV web portals (see Methods). These portals are used all over the country—and almost exclusively by PV adopters. Thus, it was not feasible gathering a non-adopters group in the same way. Unfortunately, there was no other possibility to gather a comparable non-adopter sample because it would have needed to involve persons from several areas with varying PV diffusion. Future studies should consider designs involving comparisons of adopters and non-adopters but it might be useful to concentrate on certain limited areas in this case.

The sample we investigated was not representative, especially in terms of gender, income, and education of the German population. However, the sample structure is in line with the innovators and early adopters described in the DOI theory [21] and other PV samples [41,42]. These findings are not surprising as these groups are the first to adopt innovations. It might be worthwhile doing more research, probably with other innovative technologies that have already spread further through society. Samples that are more representative could be gathered allowing the comparison between the groups defined in the DOI. It is however unlikely that those social influences are less relevant in these groups. The DOI suggests that late adopters tend to be influenced by early adopting groups who are more likely to be opinion leaders [21] and can serve as role models for later adopters [44]. Some evidence pointing in this direction was already found for PV adopters' everyday consumption behavior [45]. Indeed, subjective norms were found to predict everyday energy consumption in PV households that purchased their system at a later stage of diffusion.

Additionally, there is room for improvement in the measurements we used. For the survey measurements of social influences and the other motives, we used only two items. That limited number was inevitable as the measurement was part of a larger survey with limited space. In future research, a higher number of items could be used for calculating the scales. In the observations, we only considered PV systems in the participants' living area but not solar thermal systems as only data for PV were available. Although both systems look different (e.g., different panels and most

PV systems are considerably bigger than solar thermal systems), some nonprofessionals may have trouble distinguishing them. For future research in this area, it may be useful considering solar thermal systems as well to see if they also contribute to social influences fostering PV investments.

It should also be noted that the observation of zip code areas may involve some shortcomings. For one thing, different areas may not be entirely comparable. Some might be better suited for PV (e.g., in terms of sunshine hours per day), some may differ in social demographic issues (e.g., income or number of owner-occupied houses), and some might have experienced stronger PV marketing. Such factors may influence PV adoption and should be considered in further studies. It could also be questioned whether zip code areas provide a sufficient resolution to measure social influences. If the area was too large, decision-makers could have trouble to overview the number of PV systems within, and they might also be influenced by systems in bordering zip code areas. Another, probably higher resolution level providing more detailed information (e.g., streets or quarters) would have been desirable—but such data are hardly accessible. Even if they could be gathered, processing them would require extreme effort and involve privacy problems. Additionally, resolution issues might not be too grave at all. For one thing, zip code areas in our analysis were not extremely large. On average, they covered eight square kilometers. Additionally, zip code areas have proven a suitable indicator of social influences beforehand [7,31].

## 6. Conclusions

Our research suggests that social influences play some role in energy-relevant investment decisions. Their specific relevance is unclear as it strongly depends on the measurement approach. This finding has several implications mostly from a scientific but also from a practical viewpoint.

Several researchers claim that multiple or mixed methods should be used as one important step to improve research quality in social sciences. (see [46] for a recent overview). Research focusing on social influences is one area where such approaches are most advisable. Researchers working in this field should be well aware of the constructs (e.g., injunctive vs. descriptive norms), and methods (e.g., survey vs. observation data) that might lead to different results. Studies should always involve several measurement approaches that are compared to each other. Survey data should always cover various constructs. Such measurements should involve descriptive and injunctive norms but also, for instance, interactions with others (e.g., recommendations) prior to the investment decisions. Such determinants could also be measured in qualitative approaches (e.g., [29]) or experimental designs (e.g., [34,35]). Secondary (e.g., observation) data should be also be gathered if possible. In some cases, such data acquisition may cost some effort, but it might be the only way to get a clearer picture of the decision-relevance of social influences.

In our view, the next step is to better integrate different data sources on energy-relevant investments—namely survey and observation data. Most surprisingly, we found survey data on descriptive norms and observation data on PV installations to be uncorrelated. This finding should be investigated in further analyses. Larger samples and comparisons between adopters and non-adopters would be helpful in order to draw better conclusions.

Further studies should also look into other energy-relevant investment decisions and how they are affected by social influences. Similar relationships can be expected for those investments which are as or even more visible than PV investments, such as e-car purchases. Other energy-relevant investment, such as in-home insulations, happen more in private though. Here, social influences might be somewhat less relevant than for the visible investments.

There is also more research needed focusing on other factors than social influences. As we stated before, most of the available studies on energy-relevant behavior are concerned with rather low-impact everyday consumption—and most action models have mostly been verified for these kinds of behavior. We actually cannot really know whether these models also apply for (high-impact) energy-relevant investments but comparative studies indicate that there are some differences [5]. Rather explorative

approaches might be helpful to investigate which factors are relevant and how they interact in order to build a comprehensive conceptual framework explaining energy-relevant investment decisions.

The obvious practical implication of our research is that social influences should be better integrated into policy measures fostering pro-environmental energy-relevant investments. Most policy measures in this area focus on financial aspects where, for instance, funding and low-rate credits are provided [47]. In addition, some countries also offer professional energy consulting. Energy consulting might always involve some social influences as it generally goes along with personal contact and recommendations. Some analyses already verify the positive effects of energy consulting [34]; expanding activities in this area might be a promising approach to foster energy-relevant investments. However, energy consultants might not be the best source of social influences as they are usually strangers to the decision makers. Persons who are more familiar and/or in a similar situation (e.g., friends or neighbors) might be more suitable sources of influence. The use of a block leader would be worth considering (e.g., [27]): persons who already made an energy-relevant investment could be asked to demonstrate it to other interested persons from their neighborhood—probably against compensation. This way, any energy-relevant investment could be presented, and social influences might emerge for those that are less visible than PV use. Such an approach may cost some effort, but social influences and positive investment effects are most likely to occur. Examples supporting this idea can be found in some recent studies: Wolske et al. [30] found that the interest in talking to an installer was predicted by perceived social support and curiosity about others' PV systems; Jager [48] found a positive effect of social networks.

Finally, longitudinal studies might help make policy measures more effective. It should be investigated whether the importance of social influences and other investment determinants truly change by the time an innovation spreads through society—as theory suggests. If so, it might be promising to adopt policy measures' foci over time.

**Author Contributions:** Conceptualization, I.K.; Methodology, I.K. and I.W.; Formal Analysis, I.K.; Investigation, I.W.; Writing–Original Draft Preparation, I.K. and I.W.; Writing–Review & Editing, I.K.; Project Administration, I.W.

**Funding:** This study was conducted as part of a project within the Helmholtz-Alliance ENERGY-TRANS, funded by the Helmholtz-Association and the state of Saxony-Anhalt.

**Conflicts of Interest:** The authors declare no conflicts of interest.

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
