# Peer review of "How Measurements “Affect” the Importance of Social Influences on Household’s Photovoltaic Adoption—A German Case Study"

_sustainability, doi:10.3390/su11195175_

Round 1

Reviewer 1 Report

Very detailed and relevant work. 

Author Response

Thank you for your feedback.

A spell check will be conducted.

Reviewer 2 Report

The document contains some typos and requires a review of the language.

It is not very clear how the survey questions are composed and why the authors chose only PV for their analysis.

Authors can improve their analysis considering other "social product" in the survey.

Author Response

Thank you for your feedback.

In the new version of the manuscript the method section was extended. Now it is also pointed out more clearly why we could not investigate other energy-relevant investments that could be affected by social norms. The actual reason is there was no room for further question as the analyses was part of a much greater one referring to PV. The matter that further analyses of other social products might be most useful to validate our findings is also discussed in the conclusion section now. Here, we also discussed that other energy-relevant investments might be less affected by social influences - especially if they were less visible (e.g., retrofit measures).

We conducted another language check.

Reviewer 3 Report

Referee Report on “How Measurements “affect” the Importance of Social Influences on Household’s PV Adoption – A German Case Study”.

Manuscript number Sustainability _579057

This study uses questionnaires to explore the investment determinants on residential photovoltaic (PV) in Germany. The survey covered social influences in terms of injunctive and descriptive norm, and economic, ecological and autarky motives for the investment. Descriptive norms were more relevant for the investment decisions than injunctive norms, but both were considerably less important than all other three investment motives.

It is an interesting paper. The authors claim that the social impact of energy facility procurement is underestimated. However, I have the following specific concerns.

Major Concerns and Comments:

1. The authors mention some of the theories of consumer behavior research, such as: TPB, NAM, DOI, the Stage Model of Self-Regulated Behavioral Change and the Comprehensive Action Determination Model. But did not propose their own theoretical construction. For the injunctive norm, descriptive norm and three motives, how to influence the structure of consumer decision-making is unclear. It is suggested that the author can clearly describe the theoretical structure of the evidence using graphics and text.

2. In line 262, the authors claim that all correlations were statistically significant showing moderate to strong relationships (descriptive norm: r = .51; injunctive norm: r = .69; economic motives: r = .54; ecological motives: r = .75; autarky motives: r = .64). The authors do not explain the meaning of r. I think it represents the correlation coefficient. However, the correlation coefficient between the variables is not explained by the author. 

3. Comparing Tables 1 and 3, the mean of the descriptive norm is less than the three motives. Why is the descriptive norm significantly higher than the three motives by using the paired t-tests for the mean comparisons, I think the author should calculate again.

Final Evaluation:

For the above reasons, I believe that the current situation in this article is not suitable for publication in this journal.

Author Response

Thank you for your detailed feedback.

We mainly wanted to compare how different indicators of social influences affected energy-relevant investment decision - and how far there effects were similar to those of other investment motives. However, our analyses was not based on a certain model and we did not have a conceptual framework in mind. The introduction of different models was rather meant to point out that social scientist widely agree that social influences play some role for environmentally relevant actions. As we also stated in the beginning of section 2.3 most research we have is not concerned with energy-relevant investments but everyday behavior. Thus, it is hard to say whether the current models are actually valid for energy-relevant investments (be it in PV or other domains). Our research might help building such a model but - at least in the area of investments - we are still in an explorative phase. In our view, research on other factors (e.g., attitudes, norms) is necessary before such a framework could be build for energy-relevant investments. We now repeat and discuss the matter in more detail in the conclusion section.

We added the information that "r" is referring to Pearson's Correlational Coefficient.

We are afraid that the presentation of the mean differences in table 3 was unclear. The first motive in each block is the "basic" motive and the following motives (i.e. their means) were subtracted from it. We reworked table 3 to clarify.

Round 2

Reviewer 3 Report

The authors made some corrections to the first review, and the reviewer confirmed this. I suggest that this article can be published.